# Prevalence of erectile dysfunction in Thai scleroderma patients and associated factors

**Wichien Sirithanaphol**[1], **Ajanee Mahakkanukrauh**[2☉], **Ratanavadee Nanagara**[2☉],
**Chingching Foocharoen**[ID][2☉]*

**1** Department of Surgery, Faculty of Medicine, Khon Kaen University, Khon Kaen, Thailand, **2** Department of Medicine, Faculty of Medicine, Khon Kaen University, Khon Kaen, Thailand

☉ These authors contributed equally to this work.
* fching@kku.ac.th

**Data Availability Statement:** All relevant data are within the paper and its Supporting Information files.

## Abstract

### Background

Erectile dysfunction (ED) has been reported among patients with systemic sclerosis (SSc) and primarily limited cutaneous SSc in Caucasians. While there is no data on ED among Thais in whom the diffuse cutaneous SSc subset is common.

### Objectives

We aimed to estimate the prevalence of ED among Thais with SSc, evaluate its severity, and determine the associated factors.

### Methods

We did a cross-sectional study among adult Thai male SSc patients. All eligible patients: a) completed the IIEF-15 questionnaire by themselves; b) underwent a genital examination by an experienced urologist to evaluate skin tightness of the penis, scrotum, and phimosis; and, c) were evaluated for Erection Hardness Score.

### Results

A total of 60 male SSc patients were included. The respective mean age and median disease duration was 54.8±7.2 years and 3.1 years (IQR 1.2–7.2). The definition of ED was fulfilled in 53 cases for a prevalence of 88.3% (95%CI 77.4–95.2), while 65% had severe ED, and none had skin tightness of the genitalia. Eight cases had acquired phimosis, and all were in the ED group. The patients with ED vs. those without ED had significantly lower scores for orgasm, sexual desire, and intercourse satisfaction, and trended to be older, have more severe skin tightness and have higher BMI.

### Conclusion

ED is a common problem in men with SSc and is mainly categorized as severe. The severity of SSc might increase the risk of developing ED. We found phimosis was a common genital abnormality co-occurring with ED in SSc.

**Funding:** The authors received no specific funding for this work.

**Competing interests:** The authors have declared that no competing interests exist.

## Introduction

Systemic sclerosis (SSc) is a systemic connective tissue disease that can involve vasculature. The vascular alterations and endothelial damage lead to progressive and widespread dysfunction of various internal organs [1]. Common vasculopathy in SSc includes pulmonary arterial hypertension (PAH), Raynaud's phenomenon, and digital ulcers. Erectile dysfunction (ED) is also a common vasculopathy in men with SSc; however, it might be underestimated because some patients do not seek treatment out of embarrassment.

ED is the inability to attain or maintain an erection sufficient for satisfactory sexual intercourse. ED is common in the general population, affecting around 30 million men in the USA. ED is often undertreated as less than 20% of men with ED seek treatment, even though 8 of 10 men want their doctors to ask them about ED [2]. ED was first described in SSc patients by Lally and Jimenez in 1981 [3]. Since the prevalence of sexually active male SSc patients is as high as 80% [4], ED becomes an important issue. A simple screening tool for ED is the International Index of Erectile Function-5 (IIEF-5)—a self-administered questionnaire validated in several languages with high reliability [5]. The IIEF-5 has five grades of severity: no ED (score 22–25), mild ED (sores 17–21), mild to moderate ED (scores 12–16), moderate ED (scores 8–11), and severe ED (scores 5–7). The 15-question international index of erectile function (IIEF-15) was adapted as a screening tool for ED. The questionnaire is a validated multidimensional investigation valuable for clinical assessment and treatment outcomes of ED in clinical trials [6]. A score of 0–5 is given to each of the 15 questions that examine the four domains of male sexual function: erectile function, orgasmic function, sexual desire, and intercourse satisfaction [6]. The score of the erectile function domain based on the IIEF-15 questionnaire is also categorized into five grades: no ED (score 26–30), mild ED (score 22–25), mild to moderate ED (score 17–21), moderate ED (score 11–16), and severe ED (score 6–10). A score of 1–5 means the patients did not make an effort to have sexual activity or sexual intercourse, so patients with an IIEF-15 questionnaire score of 1–10 are categorized as having severe ED [7]. The IIEF-5 and IIEF-15 are commonly used as diagnostic tools for ED and assessing ED treatment outcomes in clinical trials.

In the prospective multicentre study of the European Scleroderma Trials and Research (EUSTAR) database, 130 men with SSc were assessed for ED using the International Index for Erectile Function-5 (IIEF-5) questionnaire [4]. The majority of men with SSc (82.3%) reported ED using the IIEF-5 questionnaire, and 31% were defined as having severe ED with a median time of 4 years from the onset of the first SSc symptoms to the onset of ED [4]. ED is somewhat a relatively early complication of SSc. The study also revealed that severe skin involvement, elevated pulmonary arterial pressures, and the presence of restrictive lung disease and muscular or renal involvement were factors associated with ED in SSc [4].

ED can be psychologically debilitating and associated with low physical and emotional satisfaction and poor general happiness. ED is a clinical challenge, especially in SSc patients, among whom many factors could be involved. The disease can affect the quality of life; the current prognosis and treatment outcomes of SSc are improved with early detection and treatment. Although the prognosis is generally good, the outcome and treatment of ED in our setting are limited. Understanding the magnitude of the problem and its associated factors could provide male SSc patients with better care. Based on a previous report, although the prevalence of ED among males with SSc was high, most patients in the study were Caucasians with lcSSc [4], unlike Thais in whom dcSSc is more prevalent [8, 9]. Due to the clinical differences between Caucasian and Thai SSc patients, our objectives were to estimate prevalence, determine severity, and determine the factors associated with ED in Thai SSc patients.

## Materials and methods

We did a cross-sectional study in adult Thai male SSc patients who were at least 18 years of age, diagnosed with SSc according to the American College of Rheumatology 1980 criteria [10] or the 2013 ACR/EULAR Classification Criteria for SSc [11], and able to read and understand Thai. We excluded patients having any active medical illnesses (i.e., heart failure functional class III-IV, pneumonia, septicemia), mobility limitations that make it difficult to have sexual activity (i.e., bedridden, hemiparalysis, paraparesis), history of psychiatric disease, neurogenic dysfunction of bowel or bladder, speech, hearing, or visual disorder, or diagnosed with overlap syndrome.

Demographic data were collected, including age, disease duration, clinical characteristics of SSc, coexisting disease, smoking status, concomitant medications, and laboratory data on the enrollment date. All eligible patients a) answered the IIEF-15 questionnaire by themselves; b) underwent genital examination by an experienced urologist to evaluate skin tightness at the genitalia (penis and scrotum) by mRSS and phimosis; and, c) were evaluated for the percentage of erection and Erection Hardness Score (EHS).

### Sample size calculation

The sample size calculation was based on this study's primary objective, which was to estimate the prevalence of ED in SSc patients. The sample size was based on the prevalence of ED in the literature (82%), a significance level of 0.05, and a precision of 90%. At least 57 men with SSc were thus included in the study.

### Operational definitions

Disease duration was counted from the date of the first non-Raynaud symptom to the date of the study. ED was evaluated using question numbers 1–5 and 15 from the IIEF-15 questionnaire and classified into five grades; no ED (score 22–25), mild ED (sore 17–21), mild to moderate ED (score 12–16), moderate ED (score 8–11), and severe ED (score 5–7) [7]. EHS was categorized into four grades: 0 –penis does not enlarge, 1 –penis is larger but not hard, 2 –penis is hard but not hard enough for penetration, 3 –penis is hard enough for penetration but not completely hard, and 4 - penis is completely hard and fully rigid [12]. The percentage of erection was the percentage of felt penile erection hardness defined by the patient (range, 0–100). Acquired or pathological phimosis was tightness of the foreskin so that it could be retracted from around the glans of the penis—a condition that was not present at birth.

### Statistical analysis

Descriptive statistics for baseline data are presented as percentages for categorical data, means ±standard deviation (SD) or medians with interquartile ranges (IQR) for continuous data. The prevalence of ED with its 95% confidence interval (95%CI) was estimated. Factors associated with ED were evaluated using one-by-one statistical tests and reported as the Odds Ratio (OR) with 95%CI. The parameters giving $p < 0.10$ were included in a logistic regression analysis. $P < 0.05$ was reported as statistically significant. The statistical analyses were done using STATA version 16.0.

The Human Research Ethics Committee of Khon Kaen University reviewed and approved the study per the Helsinki Declaration and the Good Clinical Practice Guidelines (HE601282). All eligible patients signed informed consent before entry into the study.

## Results

A total of 60 men SSc were enrolled in the study. The respective mean age and median disease duration was 54.8±7.2 years and 3.1 years (IQR 1.2–7.2). ED was defined in 53 cases for a prevalence of 88.3% (95%CI 77.4–95.2). The prevalence was 50% in patients under 40, 88% in those 40–59, 92.9% in those 60–69, and 100% in those over 70 years of age. The majority of patients with ED (33 cases; 32.3%) had an EHS score of 1, 16 (30.2%) had a score of 2, and 3 (7.6%) had a score of 3. Most ED patients (51 cases) had a percentage of erection $\leq$ 50%. Severe ED was the most common degree of ED among SSc patients (39 cases; 65%), among whom the majority (28 cases) did not attempt sexual activity. The second most common was moderate ED (9 cases; 15%), followed by mild to moderate ED (3 cases; 5%). All patients with ED had a skin score of 0 for the penis and scrotum. Phimosis was detected in 8 of 60 cases (13.3%). One had incomplete clinical data on his SSc, so was excluded from the analysis.

According to the univariate analysis, no clinical characteristics of SSc were associated with the development of ED in SSc; however, patients with ED trended to be older, have more severe skin tightness, and have higher BMI than patients without ED (p = 0.09, 0.11, 0.11 and 0.11, respectively). In addition, when the analysis included other IIEF-15 domains, patients with ED had significantly lower scores for orgasm, sexual desire, intercourse satisfaction, and overall satisfaction with erectile function than patients without ED (p<0.001 all domains). Overall clinical characteristics and the clinical comparison between patients with and without ED are presented in Table 1. The IIEF-15 scores in other domains classified by ED are presented in Fig 1.

When categorizing the severity of ED, there were no clinical differences between degrees of severity (Table 2). Notwithstanding, the patients with severe ED had significantly lower scores for orgasm, sexual desire, intercourse satisfaction, and overall satisfaction of erectile function than patients with less severity. The clinical differences categorized by severity of ED are presented in Table 2. The severity of ED in each age group is presented in Fig 2.

## Discussion

ED is a common problem in men with SSc. Our study revealed that the prevalence of ED was nearly 90% using the IIEF-15 questionnaire. The prevalence of ED in Thai men with SSc was slightly higher than in previous studies, which reported rates between 80–86% [4, 13, 14]. We also found that ED prevalence trended higher in elderly patients than in young men as in a previous study [4], although the difference was not statistically significant. In general, ED is a more frequent sexual problem in elderly men. Our study revealed that the general population has similar ED prevalence across age groups, albeit the prevalence in men over 70 with SSc was somewhat higher than the general population [15]. In general, the prevalence of ED in men under 61 years is quite low. The respective worldwide prevalence of ED in men under 40, 40–49, 60–69, and 70–80 years is between 1–10%, 2–15%, 20–40%, and 50–100% [15], while in our SSc group it was 50%, 88%, 93%, and 100%. We, however, included the population from a single centre, so the results are not generalizable. The prevalence of ED among SSc in a large population would provide more precise information about the epidemiology of ED among SSc patients.

The severity of SSc disease is associated with the risk of developing ED. A previous study revealed the association between ED and disease severity in terms of extensive skin tightness evaluated by mRSS, and having restrictive lung function, high pulmonary arterial pressure, and renal crisis or muscle involvement [4]. High mRSS and high body mass index (BMI) trended to be associated with developing ED in our SSc patients, albeit not statistically significant. In addition, our patients with a high WHO functional class, having Raynaud's

**Table 1. Overall clinical characteristics and clinical association with ED by univariate analysis.**

| Parameters | Overall | No ED | ED | OR (95%CI) | p-value |
|---|---|---|---|---|---|
| | N = 59 | N = 7 | N = 52 | | |
| Age (years); mean±SD | 54.8±7.2 | 50.4±7.2 | 55.4±7.0 | 1.10 (0.98–1.23) | 0.09 |
| Age group (N = 60) | | | | | |
| <40 years (%) | 2 (3.3) | 1 (14.3) | 1 (1.9) | 1 | |
| 40–59 years (%) | 42 (70.0) | 5 (71.4) | 37 (69.8) | 7.40 | 0.18 |
| | | | | (0.39–137.88) | |
| 60–69 years (%) | 14 (23.3) | 1 (14.3) | 13 (24.5) | 13.0 | 0.14 |
| | | | | (0.42–404.62) | |
| >70 years (%) | 2 (3.3) | 0 | 2 (3.8) | NA | - |
| Duration of disease (years); median (IQR) | 3.1 (1.2–7.2) | 4.6 (2.1–10.2) | 2.8 (0.9–7.2) | - | 0.65 |
| dcSSc subset | 43 (72.9) | 5 (71.4) | 39 (73.1) | 1.09 (0.09–7.60) | 0.93 |
| mRSS (points); median (IQR) | 7 (2–12) | 2 (0–7) | 7 (2–12) | | 0.11 |
| BMI (kg/m$^2$); mean±SD | 22.3±4.3 | 18.9±2.4 | 22.8±4.4 | 1.62 (0.90–2.91) | 0.11 |
| WHO functional class (N = 45) | | | | | |
| I (N = 26) | 26 (57.9) | 5 (83.3) | 21 (53.9) | 1 | - |
| II (N = 18) | 18 (40.0) | 1 (16.7) | 17 (43.6) | 4.04 (0.43–38.03) | 0.21 |
| III (N = 1) | 1 (2.2) | 0 | 1 (2.6) | NA | - |
| Raynaud's phenomenon (%) | 8 (13.7) | 0 | 8 (15.4) | NA | - |
| Digital ulcer (%) | 10 (17.0) | 1 (14.3) | 9 (17.3) | 1.26 (0.13–64.09) | 0.84 |
| Telangiectasia (%) | 21 (35.6) | 2 (28.6) | 19 (36.5) | 1.44 (0.29–16.41) | 0.68 |
| Calcinosis cutis (%) | 1 (1.7) | 0 | 1 (1.9) | NA | - |
| Salt and pepper skin (%) | 27 (45.8) | 3 (48.9) | 24 (46.2) | 1.14 (1.74–8.57) | 0.87 |
| Edematous skin (%) | 15 (25.4) | 1 (14.3) | 14 (26.9) | 2.21 (0.23–108.8) | 0.47 |
| Tendon friction rub (%) | 14 (23.7) | 2 (28.6) | 12 (23.1) | 0.75 (0.11–8.87) | 0.75 |
| Hand deformity (%) | 18 (30.5) | 1 (14.3) | 17 (32.7) | 2.91 (0.31–141.79) | 0.32 |
| Synovitis (%) | 6 (10.2) | 1 (14.3) | 5 (9.6) | 0.64 (0.01–35.16) | 0.70 |
| Muscle weakness (%) | 2 (3.4) | 0 | 3 (3.9) | NA | - |
| Esophageal involvement (%) | 26 (44.1) | 2 (28.6) | 24 (46.2) | 2.14 (0.31–24.14) | 0.38 |
| Stomach involvement (%) | 10 (17.0) | 0 | 10 (19.2) | NA | - |
| Intestinal involvement (%) | 4 (6.8) | 0 | 4 (7.7) | NA | - |
| Pulmonary fibrosis (%) | 33 (55.9) | 3 (42.9) | 30 (57.7) | 1.82 (0.27–13.55) | 0.46 |
| Pulmonary arterial hypertension (%) | 1 (17) | 0 | 1 (1.9) | NA | - |
| Phimosis (%) | 8 (13.3) | 0 | 8 (15.1) | NA | 0.58 |
| Other domains of IIEF-15 questionnaire | | | | | |
| Orgasm (scores); median (IQR) | 0 (0–6) | 10 (9–10) | 0 (0–3) | - | <0.001* |
| Sexual desire (scores); median (IQR) | 4 (2–7) | 7 (7–9) | 4 (2–6) | - | <0.001* |
| Intercourse satisfaction (scores); median (IQR) | 4 (2–6.5) | 9 (8–10) | 4 (2–5) | - | <0.001* |
| Overall satisfaction (scores); median (IQR) | 4.5 (2–7) | 8 (8–10) | 4 (2–6) | - | <0.001* |

*statistical significant

ED erectile dysfunction, SD standard deviation, IQR interquartile range, mRSS modified Rodnan skin score, BMI body mass index, WHO World Health Organization, IIEF-15 the International Index for Erectile Function-5, NA data not available by statistical analysis due to one cell were zero

phenomenon, muscle weakness, and/or gastrointestinal involvement more frequently presented ED than patients who did not have those conditions. Our findings cannot, however, be confirmed by a statistical test as we had no case comparisons without ED. Thai SSc patients had more disease severity than Caucasians as Thais had dcSSc more commonly than

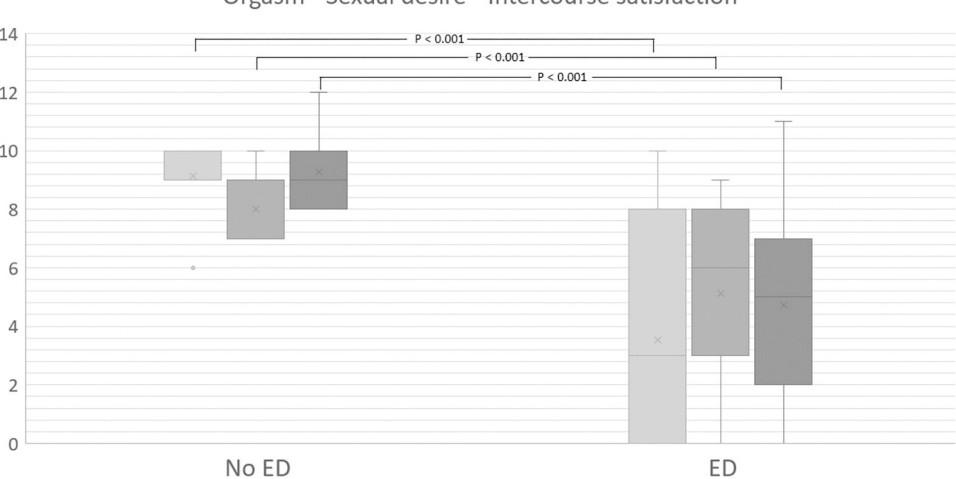

**Fig 1. IIEF-15 score in another domain classified by ED.** ED erectile dysfunction.

Caucasians while Caucasians had lcSSc more commonly than Thais, and dcSSc is a more severe disease with a poorer prognosis. Nearly 75% of our patients had dcSSc, while 53–63% had dcSSc in previous studies [4, 14], so the prevalence of ED in the severe form of SSc may be higher than the mild form.

The cause and pathogenesis of ED in SSc remain unclear. Several etiologies contribute to the development of ED, including age, medications, neurogenic, psychogenic, and vasculogenic causes [2, 16–18]. The hypothesis of the combination of fibrosis and vasculopathy in SSc has been thoroughly studied [19–21]. One study of SSc revealed diminishing systolic and diastolic blood flow velocities (evaluated by duplex ultrasound) in the penile arteries [20], and another study compared to normal controls found reduced temperature (as assessed by thermal imaging) of the penis with ED [21]. Rosato et al. found a positive correlation between the IIEF-5 score and digital sphygmic waves and skin perfusion evaluated using digital photoplethysmography. The authors also found an association between the IIEF-5 score and laser Doppler perfusion imaging of the skin and the digits [22]. In addition, female sexual dysfunction was reported, and there was a negative correlation between the Doppler indices of the clitoral artery and the Female Sexual Function Index score [23]. The findings exemplify the role of vasculopathy for ED development in male and female sexual dysfunction in SSc. Tissue fibrosis was a possible pathogenesis of ED in SSc. Nehra et al. found corporal fibrosis in penile tissues obtained during penile prosthesis insertion [19]. Sex hormone and neurological were thought to be possible roles of ED in SSc, but current data do not support the hypothesis [20, 24]. To date, the cause of ED in SSc has yet to be defined.

Phimosis might be a skin involvement of the genitalia coexisting with the development of ED in men with SSc. Although phimosis is found in the general male population of all ages, the prevalence of phimosis in our SSc patients was greater (13.3% in our study vs. 5.6% in a systematic review) [25]. There are many causes of phimosis, including trauma, infection, and inflammation [26]. According to the clinical features of SSc, skin tightness is the classical sign of SSc, so phimosis might be a skin involvement of the genitalia leading to tightness of the foreskin, making it difficult to retract. Notably, all of those patients having phimosis were in the ED group, and none of those who did not have ED developed phimosis. The causes of phimosis and the association between phimosis and ED development in SSc have never been

**Table 2. Clinical differences categorized by severity of ED.**

| Parameters | No ED | Mild ED | Mild to moderate ED | Moderate ED | Severe ED | p-value |
|---|---|---|---|---|---|---|
| | N = 7 | N = 2 | N = 3 | N = 9 | N = 38 | |
| Age (years); mean±SD | 50.4±7.2 | 55.0±4.2 | 45.0±9.5 | 58.1±5.4 | 55.6±6.8 | 0.84 |
| Age groups (N = 60) | | | | | | |
| <40 years (N = 2) (%) | 1 (14.3) | 0 | 1 (33.3) | 0 | 0 | 0.27 |
| 40–59 years (N = 42) (%) | 5 (71.4) | 2 (100.0) | 2 (66.7) | 5 (55.6) | 28 (71.8) | |
| 60–69 years (N = 14) (%) | 1 (14.3) | 0 | 0 | 4 (44.4) | 9 (23.1) | |
| >70 years (N = 2) (%) | 0 | 0 | 0 | 0 | 2 (5.1) | |
| Duration of disease (years); median (IQR) | 4.6 | 7.9 | 0.7 | 6.3 | 1.9 | 0.31 |
| | (2.1–10.2) | (6.2–9.7) | (0.3–5.5) | (3.7–14.8) | (0.9–6.5) | |
| dcSSc subset | 5 (71.4) | 2 (100.0) | 2 (66.7) | 7 (77.8) | 27 (71.1) | 0.99 |
| mRSS (points); median (IQR) | 2 (0–7) | 18 (11–25) | 10 (7–13) | 2 (0–6) | 8 (2–13) | 0.08 |
| BMI (kg/m$^2$); mean±SD | 18.9±2.4 | 18.2±2.4 | 22.8±3.2 | 22.3±2.4 | 22.8±4.7 | 0.39 |
| WHO functional class (N = 45) (%) | | | | | | 0.72 |
| I (N = 26) | 5 (83.3) | 1 (100.0) | 2 (66.7) | 4 (66.7) | 14 (48.3) | |
| II (N = 18) | 1 (16.7) | 0 | 1 (33.3) | 2 (33.3) | 14 (48.3) | |
| III (N = 1) | 0 | 0 | 0 | 0 | 1 (3.5) | |
| Raynaud's phenomenon (%) | 0 | 0 | 1 (33.3) | 2 (22.2) | 5 (13.2) | 0.48 |
| Digital ulcer (%) | 1 (14.3) | 1 (50.0) | 0 | 0 | (21.1) | 0.33 |
| Telangiectasia (%) | 2 (28.6) | 1 (50.0) | 2 (66.7) | 4 (44.4) | 12 (31.6) | 0.69 |
| Calcinosis cutis (%) | 0 | 0 | 0 | 0 | 1 (2.6) | 0.99 |
| Salt and pepper skin (%) | 3 (48.9) | 1 (50.0) | 1 (33.3) | 2 (22.2) | 20 (52.6) | 0.56 |
| Edematous skin (%) | 1 (14.3) | 0 | 2 (66.7) | 1 (11.1) | 11 (29.0) | 0.36 |
| Tendon friction rub (%) | 2 (28.6) | 2 (100.0) | 1 (33.3) | 1 (11.1) | 8 (21.1) | 0.12 |
| Hand deformity (%) | 1 (14.3) | 0 | 0 | 2 (22.2) | 15 (39.5) | 0.45 |
| Synovitis (%) | 1 (14.3) | 0 | 0 | 1 (11.1) | 4 (10.5) | 0.99 |
| Muscle weakness (%) | 0 | 0 | 0 | 0 | 2 (5.3) | 0.99 |
| Esophageal involvement (%) | 2 (28.6) | 1 (50.0) | 1 (33.3) | 3 (33.3) | 19 (50.3) | 0.78 |
| Stomach involvement (%) | 0 | 0 | 1 (33.3) | 0 | 9 (23.7) | 0.24 |
| Intestinal involvement (%) | 0 | 1 (50.0) | 0 | 1 (11.1) | 2 (5.3) | 0.19 |
| Pulmonary fibrosis (%) | 3 (42.9) | 1 (50.0) | 2 (66.7) | 5 (55.7) | 22 (57.9) | 0.97 |
| Pulmonary arterial hypertension (%) | 0 | 0 | 0 | 0 | 1 (2.6) | 0.99 |
| Other domains of IIEF-15 questionnaire | | | | | | |
| Orgasm (scores); median (IQR) | 10 (9–10) | 9 (8–10) | 7 (4–8) | 5 (2–6) | 0 (0–0) | 0.02[1]* |
| | | | | | | <0.001[2]* |
| | | | | | | 0.004[3]* |
| | | | | | | 0.01[4]* |
| | | | | | | 0.01[5]* |
| Sexual desire (scores); median (IQR) | 7 (7–9) | 7.5 (6–9) | 7 (7–8) | 4 (4–8) | 3 (1–5) | 0.04[1]* |
| | | | | | | <0.001[2]* |
| | | | | | | 0.03[3]* |
| | | | | | | 0.01[4]* |
| | | | | | | 0.03[5]* |
| Intercourse satisfaction (scores); median (IQR) | 9 (8–10) | 9 (7–11) | 7 (7–7) | 6 (5–6) | 3 (1–4) | 0.04[1]* |
| | | | | | | <0.001[2]* |
| | | | | | | 0.01[3]* |
| | | | | | | 0.005[4]* |
| | | | | | | 0.002[5]* |

*(Continued)*

**Table 2.** (Continued)

| Parameters | No ED | Mild ED | Mild to moderate ED | Moderate ED | Severe ED | p-value |
|---|---|---|---|---|---|---|
| | N = 7 | N = 2 | N = 3 | N = 9 | N = 38 | |
| Overall satisfaction (scores); median (IQR) | 8 (8–10) | 6.5 (6–7) | 8 (4–8) | 6 (5–8) | 4 (2–6) | 0.04[1*] |
| | | | | | | <0.001[2*] |
| | | | | | | 0.03[4*] |
| | | | | | | 0.01[5*] |

*statistical significant

[1]comparison between no ED and moderate ED

[2]comparison between no ED and severe ED

[3]comparison between mild ED and severe ED

[4]comparison between mild to moderate ED and severe ED

[5]comparison between moderate ED and severe ED

ED erectile dysfunction, SD standard deviation, IQR interquartile range, mRSS modified Rodnan skin score, BMI body mass index, WHO World Health Organization, IIEF-15 the International Index for Erectile Function-5, NA data not available by statistical analysis due to one cell were zero

described. We suggest a further case control study in a large population to determine the prevalence of phimosis among SSc patients and its associated factors. A foreskin biopsy might be performed to help identify the causes of phimosis.

Our study has some limitations: a) we had no control to make a comparison of the prevalence between the general population and SSc patients; b) the study was conducted at a single centre, so the results might not be generalizable; c) there were some missing data such as the WHO functional class; d) there was a small number of patients with no ED that might have resulted in a low power of analysis; and, e) multicollinearity of the data was not investigated. Notwithstanding, our study included several parameters such as EHS, skin tightness score of the genitalia, and domains of sexual function other than erectile function that has never before been recorded in male SSc patients. Hence, the findings provide new information on erectile function in such patients.

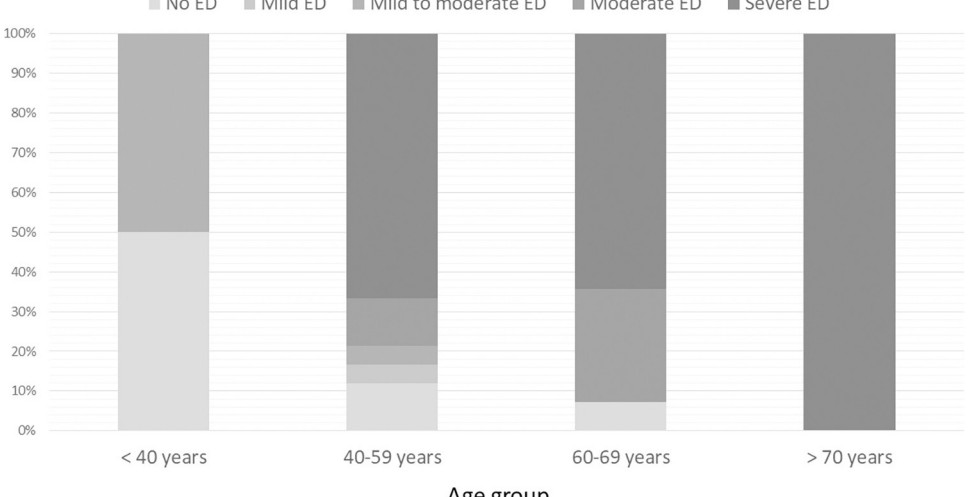

**Fig 2. Severity of ED classified by age group.** ED erectile dysfunction.

## Conclusion

ED is a common problem in men with SSc, and the prevalence increases with age. The majority of patients were categorized as having severe ED. The severity of SSc might increase the risk of developing ED. Phimosis was a common genital abnormality that coexisted with ED in SSc.

## Supporting information

**S1 File.**
(XLSX)

## Acknowledgments

The authors thank (a) the Scleroderma Research Group, the Research and Graduate Studies, and the Faculty of Medicine, Khon Kaen University, for their support, and (b) Mr. Bryan Roderick Hamman for assistance with the English-language presentation under the aegis of the Publication Clinic Khon Kaen University, Thailand.

## Author Contributions

**Conceptualization:** Wichien Sirithanaphol, Chingching Foocharoen.

**Data curation:** Ajanee Mahakkanukrauh, Ratanavadee Nanagara.

**Formal analysis:** Chingching Foocharoen.

**Funding acquisition:** Wichien Sirithanaphol.

**Investigation:** Wichien Sirithanaphol.

**Methodology:** Chingching Foocharoen.

**Supervision:** Ajanee Mahakkanukrauh, Ratanavadee Nanagara, Chingching Foocharoen.

**Visualization:** Ajanee Mahakkanukrauh, Ratanavadee Nanagara.

**Writing – original draft:** Wichien Sirithanaphol.

**Writing – review & editing:** Chingching Foocharoen.

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
