## [Decision Letter · Decision Letter 0]

8 Aug 2022

PONE-D-21-36106Prevalence of erectile dysfunction in Thai scleroderma patients and associated factorsPLOS ONE

Dear Dr. Foocharoen,

Thank you for submitting your manuscript to PLOS ONE. After careful consideration, we feel that it has merit but does not fully meet PLOS ONE’s publication criteria as it currently stands. Therefore, we invite you to submit a revised version of the manuscript that addresses the points raised during the review process.

We look forward to receiving your revised manuscript.

Kind regards,

Luca Navarini

Academic Editor

PLOS ONE

Journal Requirements:

3. Thank you for stating the following in the Acknowledgments Section of your manuscript: "The authors thank (a) the Research and Graduate Studies, the Scleroderma Research Group and the Faculty of Medicine, Khon Kaen University, for their support, and (b) Mr. Bryan Roderick Hamman for assistance with the English-language presentation under the aegis of the Publication Clinic Khon Kaen University, Thailand."

Please remove any funding-related text from the manuscript and let us know how you would like to update your Funding Statement. Currently, your Funding Statement reads as follows: "Unfunded studies"

4. Thank you for stating the following in your Competing Interests section: "No authors have competing interests"

Reviewers' comments:

Reviewer's Responses to Questions

**Comments to the Author**

1. Is the manuscript technically sound, and do the data support the conclusions?

Reviewer #1: Yes

Reviewer #2: Partly

2. Has the statistical analysis been performed appropriately and rigorously? 

Reviewer #1: Yes

Reviewer #2: Yes

3. Have the authors made all data underlying the findings in their manuscript fully available?

Reviewer #1: Yes

Reviewer #2: Yes

4. Is the manuscript presented in an intelligible fashion and written in standard English?

Reviewer #1: Yes

Reviewer #2: Yes

5. Review Comments to the Author

Reviewer #1: This is an interesting and well written study on “Prevalence of erectile dysfunction in Thai scleroderma patients and associated factors”. The authors suggest for the first time to study abnormal urological disease like phimosis, etc.

My minor points are:

Line 196, “A previous study 196 revealed the association between ED and disease severity in terms of extensive skin tightness 197 evaluated by mRSS, having restrictive lung function, high pulmonary arterial pressure, and renal or muscle involvement.” What is the renal involmente? Was renal scleroderma crisis relate to ED?

Loine 214: “The findings represent the role of vasculopathy for ED development in SSc” Please add tehe reference Rosato et al and discuss the role of endothelium dysfunction with microvascular damage in SSc patients with ED. (Rosato E, Barbano B, Gigante A, Aversa A, Cianci R, Molinaro I, Quarta S, Pisarri S, Afeltra A, Salsano F. Erectile dysfunction, endothelium dysfunction, and microvascular damage in patients with systemic sclerosis. J Sex Med. 2013 May;10(5):1380-8. doi: 10.1111/jsm.12110. Epub 2013 Feb 27. PMID: 23444914.)

Reviewer #2: The authors presented a well detailed and thorough assessment of sexual disfunction in male patients affected by Systemic Sclerosis, the quality of the study and the combined urological assessment make it very good sound. However, there are some points that may be implemented to make the paper even better in my opinion:

I'm following the line orders of the downloaded reviewer version

101. evaluated should be changed into "were evaluated for" (maybe just a typo)

120. I would suggest a more scientific term for "tip of the penis"

126. "Univariable analysis": I would suggest to name one by one the statistical tests that were used

126. "Odds ratio": odds ratios for continuous predictors were extimated with a one-predictor logistic regression, exponentiating the coefficient on the predictor and its confidence bounds, and then reporting this as the odds ratio and its 95% confidence interval?

I feel this part overall could be more detailed

162-164: I think that this particular data is redundant and suffers from multicollinearity, I would note it down straight afterwards or in the limitations of the study

223-224: the concept about phimosis: How was the skin thightness score in patients with phimosis? Since you said in the results that the skin score for the genitalia in all patients was zero, this assumption might be too arbitrary.

Lastly I would cite: J Scleroderma Relat Disord. 2019 Feb;4(1):71-76. doi: 10.1177/2397198318776593.

6. PLOS authors have the option to publish the peer review history of their article (what does this mean?). If published, this will include your full peer review and any attached files.

Reviewer #1: No

Reviewer #2: No

---

## [Author Response · Author response to Decision Letter 0]

11 Aug 2022

Dear Dr.Luca Navarini,

Academic Editor PLOS ONE

RE: PONE-D-21-36106

Title: Prevalence of erectile dysfunction in Thai scleroderma patients and associated factors

Thank you very much for your letter dated June 15, 2022, including the reviewers’ comments. We are grateful for the opportunity to improve our original manuscript, and for your consideration of this revised version. We have made changes to the original manuscript, addressing the reviewers’ concerns. The changes are listed below. We believe that the changes suggested by the reviewers strengthen the study, making for a greater impact on the field of scleroderma. 

Thank you very much for your consideration of this updated version.

Sincerely,

Chingching Foocharoen, MD (on behalf of all the authors)

Review Comments to the Author

Reviewer #1: This is an interesting and well written study on “Prevalence of erectile dysfunction in Thai scleroderma patients and associated factors”. The authors suggest for the first time to study abnormal urological disease like phimosis, etc.

My minor points are:

Line 196, “A previous study 196 revealed the association between ED and disease severity in terms of extensive skin tightness 197 evaluated by mRSS, having restrictive lung function, high pulmonary arterial pressure, and renal or muscle involvement.” What is the renal involvement? Was renal scleroderma crisis related to ED?

Response: Yes, renal involvement was renal crisis. We revised the sentence to make it clearer.

The severity of SSc disease is associated with the risk of developing ED. A previous study revealed the association between ED and disease severity in terms of extensive skin tightness evaluated by mRSS, and having restrictive lung function, high pulmonary arterial pressure, and renal crisis or muscle involvement.4

Line 214: “The findings represent the role of vasculopathy for ED development in SSc” Please add the reference Rosato et al and discuss the role of endothelium dysfunction with microvascular damage in SSc patients with ED. (Rosato E, Barbano B, Gigante A, Aversa A, Cianci R, Molinaro I, Quarta S, Pisarri S, Afeltra A, Salsano F. Erectile dysfunction, endothelium dysfunction, and microvascular damage in patients with systemic sclerosis. J Sex Med. 2013 May;10(5):1380-8. doi: 10.1111/jsm.12110. Epub 2013 Feb 27. PMID: 23444914.)

Response: Thank you for the suggestion. We reviewed the article and cited the article in the text as below.

Rosato et al. found a positive correlation between the IIEF-5 score and digital sphygmic waves and skin perfusion evaluated using digital photoplethysmography. The authors also found an association between the IIEF-5 score and laser Doppler perfusion imaging of the skin and the digits.22 In addition, female sexual dysfunction was reported, and there was a negative correlation between the Doppler indices of the clitoral artery and the Female Sexual Function Index score.23 The findings exemplify the role of vasculopathy for ED development in male and female sexual dysfunction in SSc. 

Reviewer #2: The authors presented a well detailed and thorough assessment of sexual disfunction in male patients affected by Systemic Sclerosis, the quality of the study and the combined urological assessment make it very good sound. However, there are some points that may be implemented to make the paper even better in my opinion:

I'm following the line orders of the downloaded reviewer version

101. evaluated should be changed into "were evaluated for" (maybe just a typo)

Response: The term has been changed according to the suggestion.

120. I would suggest a more scientific term for "tip of the penis"

Response: Thank you for the suggestion. The term has been changed to “glans of penis”

126. "Univariable analysis": I would suggest to name one by one the statistical tests that were used

Response: Thank you for the suggestion. The term has been changed to one used in statistical tests.

126. "Odds ratio": odds ratios for continuous predictors were estimated with a one-predictor logistic regression, exponentiating the coefficient on the predictor and its confidence bounds, and then reporting this as the odds ratio and its 95% confidence interval?

Response: Yes, the odds ratio, used for continuous data, was calculated by exponentiation of the coefficient from logistic regression. 

I feel this part overall could be more detailed

162-164: I think that this particular data is redundant and suffers from multicollinearity, I would note it down straight afterwards or in the limitations of the study

Response: We appreciate the comment. The multicollinearity was not investigated, so we added the statement in limitations of the study as below.

Our study has some limitations: a) we had no control to make a comparison of the prevalence between the general population and SSc patients; b) the study was conducted at a single centre, so the results might not be generalizable; c) there were some missing data such as the WHO functional class; d) there was a small number of patients with no ED that might have resulted in a low power of analysis; and, e) multicollinearity of the data was not investigated. 

223-224: the concept about phimosis: How was the skin thightness score in patients with phimosis? Since you said in the results that the skin score for the genitalia in all patients was zero, this assumption might be too arbitrary.

Response: Thank you for the comments. Phimosis is tightness of the foreskin of the penis, making the skin unable to be pulled back or retracted. The foreskin in phimosis is not thickness like the skin thickness as in SSc. In addition, skin in the other areas of genitalia (penis and scrotum) were normal, so the skin score of the genitalia were all evaluated with a score of zero, according to the method of mRSS assessment.

Lastly I would cite: J Scleroderma Relat Disord. 2019 Feb;4(1):71-76. doi: 10.1177/2397198318776593.

Response: Thank you for the suggestion. We reviewed the article and cited the article in the text as below.

The cause and pathogenesis of ED in SSc remain unclear. Several etiologies contribute to the development of ED, including age, medications, neurogenic, psychogenic, and vasculogenic causes.2,15–17 The hypothesis of the combination of fibrosis and vasculopathy in SSc have been thoroughly studied.18–20 One study of SSc revealed diminishing systolic and diastolic blood flow velocities (evaluated by duplex ultrasound) in the penile arteries,19 and another study compared to normal controls found reduced temperature (as assessed by thermal imaging) of the penis with ED.20 Rosato et al. found a positive correlation between the IIEF-5 score and digital sphygmic waves and skin perfusion evaluated using digital photoplethysmography. The authors also found an association between the IIEF-5 score and laser Doppler perfusion imaging of the skin and the digits.22 In addition, female sexual dysfunction was reported, and there was a negative correlation between the Doppler indices of the clitoral artery and the Female Sexual Function Index score.23 The findings exemplify the role of vasculopathy for ED development in male and female sexual dysfunction in SSc. Tissue fibrosis was a possible pathogenesis of ED in SSc. Nehra et al. found corporal fibrosis in penile tissues obtained during penile prosthesis insertion.18 Sex hormone and neurological were thought to be possible roles of ED in SSc, but current data do not support the hypothesis.19,23 To date, the cause of ED in SSc has yet to be defined.

---

## [Decision Letter · Decision Letter 1]

1 Dec 2022

PONE-D-21-36106R1

Prevalence of Erectile Dysfunction in Thai Scleroderma Patients and Associated Factors

PLOS ONE

Dear Dr. Foocharoen,

Thank you for submitting your manuscript to PLOS ONE. After careful consideration, we feel that it has merit but does not fully meet PLOS ONE’s publication criteria as it currently stands. Therefore, we invite you to submit a revised version of the manuscript that addresses the points raised during the review process.

We look forward to receiving your revised manuscript.

Kind regards,

Luca Navarini

Academic Editor

PLOS ONE
---

## [Editor Report · Acceptance letter]

9 Jan 2023

PONE-D-21-36106R1 

Prevalence of Erectile Dysfunction in Thai Scleroderma Patients and Associated Factors 

Dear Dr. Foocharoen:

I'm pleased to inform you that your manuscript has been deemed suitable for publication in PLOS ONE. Congratulations! Your manuscript is now with our production department. 

Kind regards, 

on behalf of

Dr. Yuan-Ti Lee 

Academic Editor

PLOS ONE